# The Moderating Role of Corporate Governance on the Associations of Internal Audit and Its Quality with the Financial Reporting Quality: The Case of Yemeni Banks

**Nabil Ahmed Mareai Senan** [1,2]

1 Department of Accounting, College of Business Administration, Prince Sattam bin Abdulaziz University, Alkharj 11942, Saudi Arabia; nabil_senan@yahoo.com
2 Accounting Department, Administrative Science College, Albaydha University, Albaydha 14517, Yemen

**Abstract:** This study investigates the moderating effect of corporate governance on the associations of the internal audit and quality of the internal audit with the quality of financial reporting among commercial banks in the Republic of Yemen. The final sample includes 210 internal auditors, heads of internal auditors, chairpersons, and members of audit committees. Using a survey-based methodology, the results of the Smart-PL4 analysis showed a positive association between the internal audit and quality of the internal audit and quality of financial reporting. Interestingly, the results showed an insignificant association between the internal audit, quality of the internal audit, and quality of financial reporting when considering the moderating effect of corporate governance. It is worth noting that the results confirm the existence of a positive relationship between the internal audit, quality of the internal audit, and quality of financial reporting. This confirms the importance of the internal audit and quality of the internal audit in enhancing the quality of financial reports and instilling confidence in improving internal control processes and the financial reporting framework. Among the study's many contributions are that it enhances current research on the interrelationship between internal auditing, quality of internal audits, and quality of financial reporting. It highlights the pivotal role of the internal audit, its effectiveness, and its ability to improve the quality of financial reports. This study calls for more stringent internal controls and posits that strengthening the internal audit and quality of the internal audit, along with improving corporate governance, can enable managers to raise financial reporting standards in banks. It also provides a mechanism for audit committees to monitor internal audit processes and evaluate internal performance.

**Keywords:** internal audit; quality of internal audit; quality of financial reporting; corporate governance; banks; Yemen; Smart PLS 4

**JEL Classification:** G21; G34; M42

## 1. Introduction

The escalating concerns over accounting fraud jeopardizing bank operations, coupled with global company failures and widespread social corruption, highlight the increasing imperative for accounting professionals to adhere to robust ethical standards (Hazaea et al. 2021). In response to past accounting mishaps, Internal Auditing (IA) is a critical element in ensuring effective controls, with its assurance and advisory responsibilities playing a vital role in supporting risk management efforts (Jarah et al. 2022). The application of a robust IA program is deemed essential for banks to monitor and oversee their operations effectively. IA serves a pivotal role in ensuring that the Accounting System (AS) encounters stakeholder demands, historically being perceived as a guarantee of financial statement accuracy and a means to ensure compliance with financial quality security standards. Despite this perception, the prevalence of fraudulent financial transactions has not significantly diminished solely due to IA efforts (Ogoun and Atagboro 2020).

Internal auditors must identify and analyze the risks of accounting errors (Drábková and Pech 2022). The effectiveness of IA is prejudiced by the audit's overall effectiveness, the experience of the auditing, and the impartiality of IA. While internal auditors are responsible for identifying financial anomalies (Betti et al. 2021), their ability to spot fraud is contingent on their education and professional experience. Notwithstanding the significance of IA, the literature often emphasizes its efficiency and overall function rather than its specific duty (Drogalas et al. 2017). In the economic climate, it has adopted various accounting alternatives, including Computer Assisted (CA) approaches, to enhance financial information and meet self-objectives related to profitability and financial status. However, the use of these approaches in financial statement preparation can potentially undermine their credibility, necessitating a thorough understanding of the processes involved and the extent to which IA can employ audits to mitigate the risks associated with these approaches.

This study aims to explore various aspects related to IA and banks' utilization of CA. The researcher seeks new data to enhance their theoretical understanding of the factors influencing the effectiveness of IA in mitigating CA practices that deviate from accepted accounting standards. The anticipated results aim to provide valuable guidance to bank departments, offering insights into the risks associated with CA and emphasizing the crucial role of IA in minimizing such accounting practices. The researcher believes that their findings will contribute to clarifying the concept of CA methods, serving as beneficial information for decision-makers and policymakers. The objective of financial reporting is to furnish clients with truthful, relevant, intelligible, comparative, timely, and verifiable financial information to facilitate decision-making. The effectiveness of financial reporting is crucial for resource allocation within a firm, as only accurate financial reporting can impact a company's ability to secure funding from external sources and ensure accountability (Murphy and O'Connell 2013). Financial reports play a pivotal role in influencing judgments by users of financial information regarding the company's likelihood of experiencing future net cash inflows and the efficiency of financial resource management by the management team (IFRS 2020). Consequently, reliable financial reporting is highly valued by various stakeholders, including shareholders, lenders, and suppliers, as it forms the basis for informed decision-making and fosters trust in the financial health and integrity of the reporting entity.

The study results confirm the existence of a positive relationship between the IA, quality of the internal audit (QIA), and quality of financial reporting (QFR). This confirms the importance of the IA and QIA in enhancing the quality of financial reports and improving internal control processes and the financial reporting framework. It highlights the pivotal role of the IA, its effectiveness, and its ability to improve the quality of financial reports. It is assumed that strengthening the IA and QIA, along with improving corporate governance (CG), can enable managers to raise financial reporting standards in banks. It also provides a mechanism for audit committees to monitor IA operations and evaluate internal performance.

Financial reports are of foremost importance in the financial market and financial companies, as reports indicate that a large percentage of organizations fail to meet the quality standards required in their financial reports. A PricewaterhouseCoopers survey in 2018 found that most reports did not meet the necessary standards.

High-profile cases, such as PwC's two-year ban in India for failing to disclose Satyam's overstated revenues, and the Securities and Exchange Commission's revelations about Hertz Global Holdings and Miller Energy, highlight the seriousness of the problem. In Uganda, financial reporting errors by commercial banks have been repeated since 1999, leading to concerns raised by researchers. For example, the value of the non-performing loans reported by Crane Bank in 2016 was much lower than the actual amount. Similar concerns about Imperial Bank's financial reporting were raised by the Bank of Uganda in 2015.

The literature suggests that factors such as CG and the QIA may impact the QFR. Effective CG is noted to enhance the oversight and management of operations. Internal

auditors improved the standard of financial reporting through their mission to oversee the processes and procedures related to the QIA. Stakeholder theory emphasizes the need for businesses to establish efficient checks and balances, including CG. However, there is a lack of clarity on the strategies that financial institutions should employ to encourage honest financial reporting. Research in emerging nations, particularly in Africa, regarding board independence, its performance, and IA effectiveness is limited. Existing studies tend to focus on non-African countries, narrowing their scope to specific sectors, such as Tier IV Microfinance Institutions in the case of Nalukenge et al. (2017). Nalukenge et al. (2018) explored the internal controls and company governance on IFRS compliance, establishing a strong connection between IFRS compliance and CG.

The authors note a scarcity of studies on financial reporting, with Nkundabanyanga et al. (2013) being an exception as they defined the QFR in the understandability of financial information. Generalizing case study outcomes, according to (Saunders et al. 2009), is deemed feasible only for individual case study organizations due to the diverse systems and backgrounds of each organization. In contrast to the single-government ministry focus (Nkundabanyanga et al. 2013), which may pose challenges in understanding findings, this study concentrates on financial institutions, predominantly those in private ownership. Conversely, Nkundabanyanga et al. (2013) broadened its scope to financial institutions. The researchers, exemplified by (Bananuka et al. 2018), assert that studies like the present one emphasize general responsibility more than financial reporting accuracy, considering accountability as an integral part of financial reporting.

Based on the above, the current study examines the moderating role of CG on the associations of the IA and its quality with the QFR: The case of Yemeni banks.

The problem of the study: what is the impact of CG on IA engagements and their quality on the quality of financial reports in the Commercial Bank of Yemen?

## 2. Literature Review and Developing Hypotheses

### 2.1. Internal Audit

An audit serves as an objective verification and advisory process that contributes to its effectiveness. The role of the IA is to assess activities for compliance, alignment with best financial practices, and effectiveness in achieving financial goals to improve financial and accounting processes (Hazaea et al. 2021). The IA also works to minimize misstatements arising from internal processes and recorded transactions. Contrary to the common perception of auditors solely focusing on fraud and errors, their primary objective is to evaluate the effectiveness of the IA (Drogalas et al. 2017). It audits and analyses financial accounting principles, procedures, and elements, providing management with objective data and recommendations for efficient utilization (Munteanu et al. 2016). The IA is considered a crucial element in developing accounting procedures to mitigate the impact of creative accounting (Ogoun and Atagboro 2020). The involvement of the IA lends greater credibility to financial statements and helps mitigate the consequences of potential issues. Internal auditors are expected to be well-versed in IA techniques, given their role in understanding audits (Saleh et al. 2023). However, adherence to industry ethics is essential for internal auditors to provide valuable insights and practical solutions to IA issues. The credit crisis highlighted by Al Momani and Obeidat (2013) increased the probability of fraud and unethical behavior, emphasizing the crucial role of auditors (Rakipi et al. 2021).

**H1.** *Posits that IA has a positive impact on QFR in the banking sector.*

### 2.2. The QIA

Roussy and Brivot (2016) worked on perceptions of the QIA that were examined among internal auditors, and members of IIA. While it defines quality as adherence to advised standards and procedures, external auditors primarily viewed the QIA. However, IA members believed the QIA was determined by the significance management attributed to IA reports. The study considered competence, independence, and dedication to professional

norms. As per Bananuka et al. (2018), an effective IA encompasses the evaluation of internal controls. The study argues that IA staff who lack professionalism, independence, and competence are not fulfilling their duties. Therefore, the hypothesis two is:

**H2.** *Posits that the QIA positively influences the QFR.*

This underscores the importance of effective IA practices in contributing to accurate and reliable financial reporting.

### 2.3. BG and SFR

Nalukenge et al. (2017) discovered a relationship between financial expertise and IC. However, the board's independence did not exhibit a similar association. In a subsequent study, Nalukenge et al. (2018) identified a robust connection between IFRS compliance and governance, specifically evaluating the effectiveness of the board function in financial literacy, independence, and governance. Mansor et al. (2013) suggested that full governance may not be a realistic option without governance. This study focuses on how governance influences accuracy and the QFR. Governance is deemed essential for financial institutions and banks to enhance productivity, management, and regulation (Changezi and Saeed 2014). Consequently, various stakeholders significantly influence the conduct and implementation. The association between a poor QFR and governance behaviors, such as financial statement fraud and earnings management, has been consistently established (Beasley and Petroni 2001). Online reporting and board functional effectiveness have also been found to be closely related (Bananuka et al. 2018, 2019a, 2019b). Studies on IFRS adoption highlighted a significant relationship between the use of IFRS in Microfinance Institutions (MFIs) and board effectiveness, incorporating financial literacy as one of the measures. Additionally, Nalukenge (2020) identified a significant relationship between IFRS disclosure standards. The above evidence underscores how traditional governance practices, including unbiased financially literate and capable directors, can elevate the standard. Hence, the hypothesis (H3) posits that bank governance has a positive impact on the QFR, emphasizing the pivotal role of governance in ensuring accurate and reliable reporting in this sector. Hypothesis three is stated below:

**H3.** *Posits that bank governance has a positive impact on the QFR.*

### 2.4. Corporate Governance, IA, and Quality of Financial Reports

The importance of bank governance (BG) has been recognized since the early 1930s, particularly with the separation of ownership and management. BG has gained global momentum due to a need for assurance and confidence, economic shocks, and the necessity to reform governance structures and norms. BG is considered a vital element of market discipline, ensuring competitive advantages for banks with effective governance systems. Effective BG practices are crucial for banks to fulfill legal requirements and fiduciary duties to investors (Levis 2006). The efficiency of BG practices also impacts the resilience of the financial system and aids in resolving disputes among shareholders, executives, and other stakeholders (Oino 2019). High-quality BG is positively related to IA. The literature shows a strong association between the efficiency of BG and IC, addressing IC flaws with stronger board management, audit committees, and senior management (Johnstone et al. 2011). The IA is recognized as a core element of BG, playing a crucial role in offering assurances and recommendations to support the supervision of the BG (Abdullah 2014). The contribution of IAF to BG is considered indisputable, especially in light of recent scandals emphasizing the need for good governance (Khlif and Samaha 2014).

**H4.** *BG moderates the relationship between the IA and QFR.*

**H5.** *BG moderates the relationship between the QIA and QFR.*

Previous studies did not talk about the moderating effect of CG on the relationship between the IA and its quality with the quality of financial reports in the Commercial Bank of Yemen. There were no previous studies at all that indicated this, and therefore the following study gap was formulated. We tried to know the direct and indirect impact of the IA and its quality on the quality of financial reports in the Commercial Bank of Yemen by using a modified CG variable. Figure 1 shows the model.

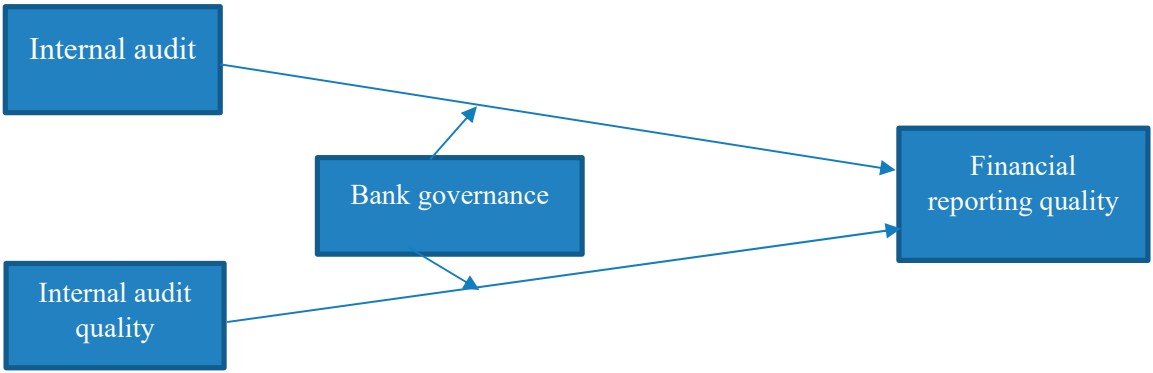

**Figure 1.** Study model.

*2.5. Methodology*

The study employed a combination of quantitative and qualitative methods, focusing on commercial banks in the Republic of Yemen. The study sample included 27 commercial banks out of 55, chosen for public attention and rigorous monitoring. Foreign exchange bureaus were excluded due to their distinct operational characteristics and vulnerability to external shocks. MFIs were also excluded from the analysis until 2019, as they were not subject to strict oversight standards. Data collection took place from February 2023 to August 2023, with 210 participants from IA departments in the target organizations responding to participant profile questionnaires. Among the respondents, 115 were senior financial officials and IA managers, with 34 women (16.2%) and 176 men (83.8%). Most participants held a university degree, including 54 with a bachelor's degree (25.6%), 61 with a master's degree (29.3%), and 95 with a doctorate degree (45.1%). Additionally, 183 respondents (87.14%) had professional certifications such as the CA program and ACCA course. Regarding age distribution, 36 participants were in the 20–30 age group (17.2%), 53 in the 31–40 age group (25.3%), and 56 in the 41–50 age group (15.0%), and 65 were 51 years or older (26.6%). Responses were collected at the individual commercial bank level, with 27 commercial banks with a response rate of 74%. The table provides additional information about the characteristics of the respondents.

The data collection for this investigation utilized a well-established questionnaire featuring closed-ended questions with a 5-point Likert scale. Participants were asked to express their agreement or disagreement with assertions, with scale points ranging from strongly disagree (1) to strongly agree (5). Closed questions were chosen for their efficiency and effectiveness for both participants and researchers. The questionnaire was distributed to academics and professionals, including auditors, accountants, and financial managers, to ensure the applicability of the questions. Expert opinions were sought, and a Content Validity Index (CVI) was calculated, resulting in a CVI of 0.90, indicating high content validity. Recommendations from both practitioners and academics were incorporated to enhance the questionnaire further. The validity of the questionnaire was assessed using Cronbach's alpha values, with coefficients for each study exceeding 0.7, indicating high reliability. The independence, effectiveness, and expertise of the board of directors were considered cornerstones of BG, based on perspectives from academics such as Nalukenge et al. (2018) and Nkundabanyanga and Ahiauzu (2012). The QIA was operationalized with employee independence, following the approaches of Roussy and Brivot (2016). Business size and age were also considered as potential influencing factors, drawing from Bartov et al. (2000).

To avoid the misinterpretation of study hypotheses, confounding factors were carefully considered, and the details of the measurement variables are provided in the table.

The reporting quality in the Republic of Yemen has been rated at 4.7, with an overall rating of 5.97. However, the overall rating of the two indicates that there are several banks with subpar financial records in the country. The questionnaires for this study were distributed to chief financial officers, chief accountants, and IA managers in the finance departments. A score of at least two suggests that issues with the financial reporting in the Republic of Yemen need to be addressed. The board's CG measures, including independence (4.2), expertise (3.7), and competence (4.3), indicate a significant dependency on these factors for effective governance. However, the scores suggest that certain organizations' boards of directors are perceived as financially illiterate and unintelligent, potentially contributing to poor financial reporting in the Republic of Yemen. The average score for the QIA was 3.9, indicating subpar performance in terms of employee ability, adherence to industry standards, and independence. This further underscores the need for improvement in IA practices. To demonstrate the data's normality, measures for skewness and kurtosis are presented. According to recommendations by Field (2009), all skewness and kurtosis values fall within an acceptable range between −3 and +3.

In the study, the independent variables include the IA, QIA, and BG, acting as moderating variables. The dependent variable is the QFR. For the measurement of the IA, six items were utilized. Similarly, six indicators were derived for the QIA. BG was assessed using six items, while eight items were proposed as indicators of the QFR. Modifications to these components were made based on (Vadasi et al. 2020). The variables and hypotheses were validated and tested using PLS-SEM. The evaluation included assessing the reliability, discriminant validity, and convergent validity of the items for each research factor. Route coefficients, R2 values, and Q2 values were also computed to analyze the model. The R2 value indicates the variance in the dependent variable. Q2 assesses the model's predictive capacity. As per the guidelines by Hair et al. (2016), 40% and 70% criteria may need to be eliminated if doing so improves the Composite Reliability (CR) or Average Variance Extracted (AVE) beyond a predefined threshold. In this study, the result was 26 retained items. The details of respondents are presented in Table 1.

**Table 1.** Study sample and respondents.

| The Study Sample | Respondents |
| --- | --- |
| Sanaa Commercial Bank | 24 |
| Dhamar Commercial Bank | 27 |
| Yarim Commercial Bank | 30 |
| Commercial Bank of Ibb | 27 |
| Commercial Bank of Taiz | 24 |
| Omran Commercial Bank | 23 |
| Al Mahwit Commercial Bank | 28 |
| Marib Commercial Bank | 27 |
| TOTAL | 210 |

The evaluation of the measurement model involved assessing the convergent and discriminant validity of the latent variables using specific items. The measurement models met the requirements based on the gathered data. The external loading was examined to assess the model, with higher external factor loadings (Hair et al. 2014). Items with an external loading greater than 0.60 were retained, resulting in 26 different items in the study. Additional assessments for data consistency, such as Cronbach's alpha and CR, were conducted to provide a more accurate measure of data consistency. ICR measures the extent to which each item on a single scale measures the same variable. For convergent validity, the AVE for each latent variable needed to be greater than 0.50, and the AVE test findings for all variables were indeed higher than 0.5, indicating a high level of internal consistency. The data presented in Table 2 and Figure 2 suggest that the model is accurate and valid. Therefore, the existing measuring strategy used in this work can be considered suitable for conducting further analyses.

**Table 2.** Factor analysis (EFA) and confirmatory factor analysis (CFA).

| | | EFA | | | CFA | | |
|---|---|---|---|---|---|---|---|
| | | PCA | | Reliability | Convergent Validity | | |
| **Construct** | **Element** | **Factor Loadings** | **% of Variance Explained by a Factor of Unidimensionality** | **Cronbach's Alpha** | **Factor Loading b** | **T Values** | ***p* Values** |
| Internal audit | | | 74.832 | 0.907 | | | |
| | IA1 | 0.827 | | | 0.826 | 37.976 | 0.000 |
| | IA2 | 0.871 | | | 0.870 | 52.603 | 0.000 |
| | IA3 | 0.859 | | | 0.859 | 46.871 | 0.000 |
| | IA4 | 0.867 | | | 0.866 | 48.943 | 0.000 |
| | IA5 | 0.788 | | | 0.787 | 27.507 | 0.000 |
| | IA6 | 0.742 | | | 0.740 | 22.597 | 0.000 |
| Quality of internal audit | | | 76.238 | 0.903 | | | |
| | QIA1 | 0.828 | | | 0.827 | 37.129 | 0.000 |
| | QIA2 | 0.853 | | | 0.852 | 44.425 | 0.000 |
| | QIA3 | 0.848 | | | 0.847 | 49.936 | 0.000 |
| | QIA4 | 0.808 | | | 0.807 | 30.336 | 0.000 |
| | QIA5 | 0.813 | | | 0.812 | 37.352 | 0.000 |
| | QIA6 | 0.776 | | | 0.775 | 27.029 | 0.000 |
| Bank governance | | | 76.500 | 0.911 | | | |
| | BG1 | 0.857 | | | 0.855 | 39.827 | 0.000 |
| | BG2 | 0.845 | | | 0.844 | 40.500 | 0.000 |
| | BG3 | 0.845 | | | 0.844 | 47.556 | 0.000 |
| | BG4 | 0.829 | | | 0.827 | 37.937 | 0.000 |
| | BG5 | 0.793 | | | 0.791 | 30.788 | 0.000 |
| | BG6 | 0.820 | | | 0.818 | 31.649 | 0.000 |
| Quality of financial reports | | | 76.628 | 0.942 | | | |
| | QFR1 | 0.822 | | | 0.820 | 35.296 | 0.000 |
| | QFR2 | 0.833 | | | 0.832 | 39.237 | 0.000 |
| | QFR3 | 0.835 | | | 0.834 | 35.089 | 0.000 |
| | QFR4 | 0.838 | | | 0.837 | 36.702 | 0.000 |
| | QFR5 | 0.876 | | | 0.875 | 53.095 | 0.000 |
| | QFR6 | 0.868 | | | 0.867 | 53.360 | 0.000 |
| | QFR7 | 0.856 | | | 0.855 | 45.290 | 0.000 |
| | QFR8 | 0.814 | | | 0.812 | 33.582 | 0.000 |

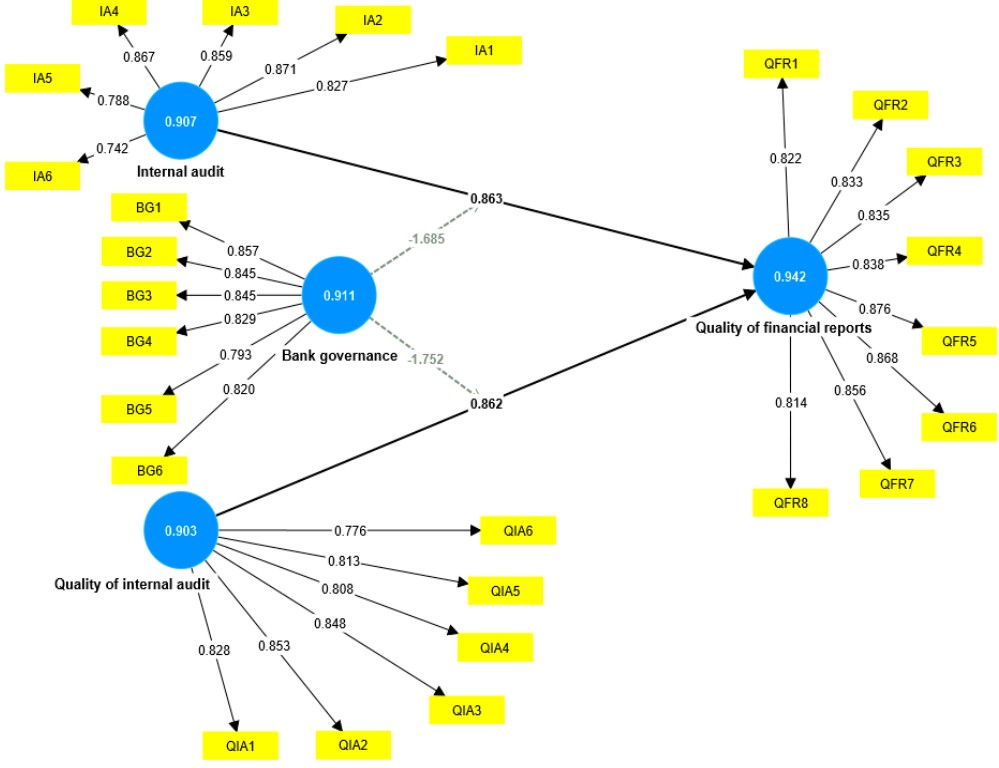

**Figure 2.** The PLS algorithm of the measurement model.

The visual representation of the structural model illustrates the relationships and the degree of influence between the IA, QIA, BG, and QFR. Additionally, the diagram illustrates the moderating effects of a control-oriented culture, highlighting the importance of a supportive culture within banks for a strong association between the IA, QIA, BG, and QFR. The results indicate that there needs to be a cultural environment within banks that aligns with these objectives to establish a robust connection between the IA, QIA, BG, and QFR. Furthermore, the findings suggest that a BG that values the IA, QIA, BG, and QFR enhances the IA, QIA, and BG. This emphasizes the integral role of a supportive governance structure in promoting effective IA practices and ensuring the QFR. The cross-loading of the constructs was examined to assess if any item loaded more heavily on its associated construct. The results table provides insights into how consistently each item loaded on its respective construct compared to other iterations, demonstrating the precision of the measurement approach used in the study. Table 3 shows cross loading.

**Table 3.** Cross loadings.

| Study Variables | | IA | QIA | BG | QFR |
|---|---|---|---|---|---|
| Internal audit | IA1 | 0.827 | 0.686 | 0.668 | 0.775 |
| | IA2 | 0.871 | 0.692 | 0.670 | 0.700 |
| | IA3 | 0.859 | 0.724 | 0.708 | 0.753 |
| | IA4 | 0.867 | 0.764 | 0.710 | 0.772 |
| | IA5 | 0.788 | 0.646 | 0.637 | 0.669 |
| | IA6 | 0.742 | 0.637 | 0.659 | 0.701 |
| Quality of internal audit | QIA1 | 0.728 | 0.828 | 0.757 | 0.764 |
| | QIA2 | 0.707 | 0.853 | 0.726 | 0.731 |
| | QIA3 | 0.704 | 0.848 | 0.699 | 0.703 |
| | QIA4 | 0.660 | 0.808 | 0.648 | 0.678 |
| | QIA5 | 0.647 | 0.813 | 0.662 | 0.686 |
| | QIA6 | 0.684 | 0.776 | 0.675 | 0.678 |
| Bank governance | BG1 | 0.668 | 0.725 | 0.857 | 0.742 |
| | BG2 | 0.650 | 0.677 | 0.845 | 0.731 |
| | BG3 | 0.704 | 0.681 | 0.845 | 0.737 |
| | BG4 | 0.656 | 0.674 | 0.829 | 0.733 |
| | BG5 | 0.686 | 0.710 | 0.793 | 0.719 |
| | BG6 | 0.714 | 0.755 | 0.820 | 0.788 |
| Quality of financial reports | QFR1 | 0.744 | 0.717 | 0.777 | 0.822 |
| | QFR2 | 0.725 | 0.716 | 0.747 | 0.833 |
| | QFR3 | 0.703 | 0.721 | 0.736 | 0.835 |
| | QFR4 | 0.749 | 0.738 | 0.762 | 0.838 |
| | QFR5 | 0.760 | 0.763 | 0.782 | 0.876 |
| | QFR6 | 0.739 | 0.739 | 0.765 | 0.868 |
| | QFR7 | 0.713 | 0.714 | 0.722 | 0.856 |
| | QFR8 | 0.684 | 0.701 | 0.725 | 0.814 |

Table 4 provides measures of internal consistency to ensure the reliability of items. Composite reliability, which assesses the internal consistency of components, was examined in this study. The composite reliability values (rho_a) for each construct fell within the recommended ranges, indicating strong internal consistency. Additionally, consistency in item connectivity was ensured by evaluating composite reliability values (rho_c), which also demonstrated high internal consistency. Cronbach's alpha, another test of internal consistency, was employed in this study. The Cronbach's alpha values for each construct exceeded the threshold value, reinforcing the internal consistency of the items within each construct. Convergent validity, which assesses the measurement of the same construct and its association with other variables, was confirmed through the calculation of AVE. Each construct in the study exceeded the AVE threshold of 0.5, indicating a satisfactory level of convergence. The Figure 3 illustrates the assessment process. Discriminant validity,

which examines the extent to which one construct differs from others, was evaluated using the Fornell–Larcker criterion and the HTMT criteria. The study ensured that discriminant validity was established based on these criteria.

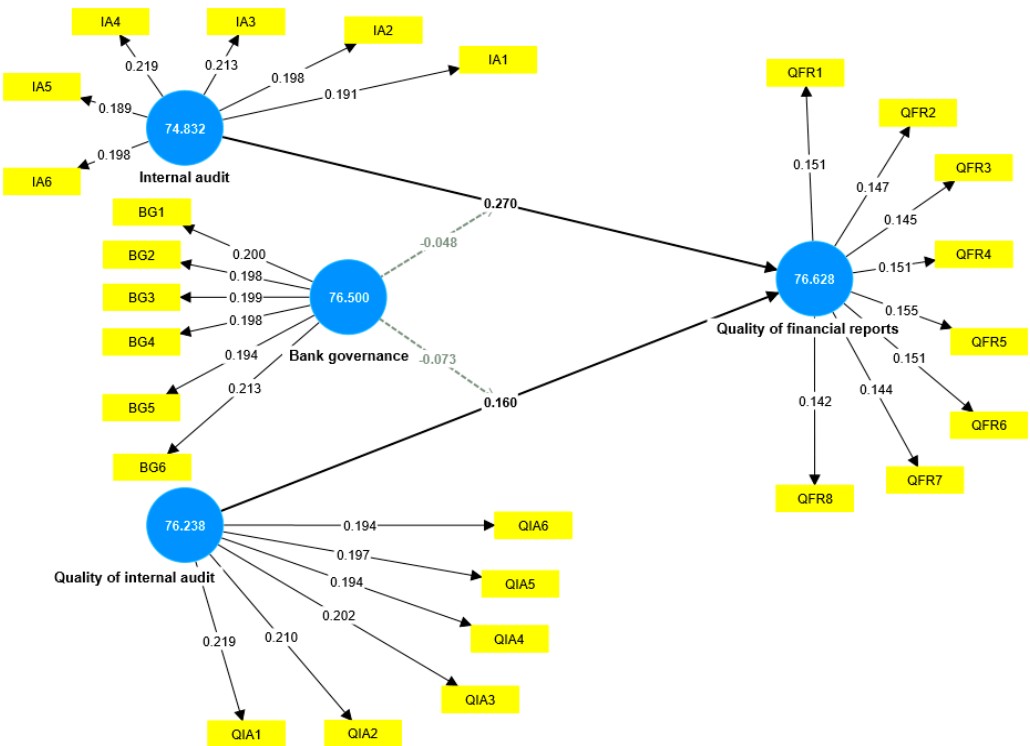

**Figure 3.** The PLS algorithm of the measurement model. % of variance explained by a factor of uni-dimensionality. The second model for analyzing A Smart Plus.

**Table 4.** Construct reliability and validity.

|  | Cronbach's Alpha | rho_a | rho_c | AVE |
|---|---|---|---|---|
| Internal audit | 0.907 | 0.908 | 0.928 | 0.684 |
| Quality of internal audit | 0.903 | 0.905 | 0.926 | 0.675 |
| Bank governance | 0.911 | 0.911 | 0.931 | 0.692 |
| Quality of financial reports | 0.942 | 0.942 | 0.952 | 0.711 |

The AVE values are higher than 0.5, with values exceeding 0.70. According to Pavlou and Fygenson (2006), these results indicate good convergent validity for the measurement model. Discriminant validity, which assesses the ability of items to differentiate between constructs, was also examined. The study considered the correlations between constructs and ensured that the square root of the AVE for each construct was substantially larger than the correlation of that construct with other constructs. The table illustrates the necessary discriminant validity of the measurement model based on these criteria. The cross-loadings of the various structures were also analyzed to determine if any particular structure was more heavily loaded than others on its associated constructs. According to Table 5, each item loads more on its own construct than on other constructs, further demonstrating the good discrimination of the measurement model.

**Table 5.** Discriminant validity.

| Bank Governance | BG | IA | QFR | QIA | BG × IA | BG × QIA |
|---|---|---|---|---|---|---|
| IA | 0.899 | | | | | |
| QFR | 0.963 | 0.932 | | | | |
| QIA | 0.931 | 0.924 | 0.933 | | | |
| BG × IA | 0.852 | 0.792 | 0.868 | 0.817 | | |
| BG × QIA | 0.870 | 0.794 | 0.877 | 0.825 | 0.975 | |

Table 6 presents the results of the Fornell–Larcker discriminant validity criterion. The values in bold type indicate the square roots of AVEs on diagonals that are greater than the correlations between constructs (represented by the corresponding row and column values). The variables demonstrate better discriminant validity when the square roots of AVEs are larger than the correlations with other constructs. According to (Awang et al. 2015), discriminant validity is satisfactory when the correlation between exogenous components is less than 0.90. Therefore, the findings in Table 6 suggest that each construct in the study exhibits satisfactory discriminant validity.

**Table 6.** Fornell–Larcker criterion.

| | BG | IA | QFR | QIA |
|---|---|---|---|---|
| BG | 0.832 | | | |
| IA | 0.818 | 0.827 | | |
| QFR | 0.893 | 0.863 | 0.843 | |
| QIA | 0.847 | 0.839 | 0.862 | 0.822 |

Table 7 and Figures 4 and 5 show the path coefficients. The proposed study hypothesis (H1) is that the IA positively affects the QFR. The results showed a positive and significant relationship between banks' IA and the QFR (b = 0.270, T = 5.869, $p > 0.000$), and hypothesis H1 was supported in this study. Hypothesis H2 is that the QIA positively affects the QFR. The results (b = 0.160, T = 3.077, $p > 0.000$) indicate that there is a positive and significant relationship between the QIA and the QFR, and hypothesis H2 was accepted. Hypothesis H3 proposes that BG positively affects the QFR. The results showed (b = 0.332, T = 6.227, $p > 0.000$), that is, BG is positively related to the QFR, and in this study, the proposed third hypothesis was accepted. The proposed hypothesis H4 is that BG works to modify the relationship between the IA and the QFR. The results show (b = −0.048, T = 0.868, $p < 0.000$), meaning that BG does not modify the relationship between the IA and the QFR, as the value of (b) is less than 0 and the value of (T) is less than 2, so the fourth hypothesis was rejected. Hypothesis H5: BG moderates the relationship between the QIA and QFR. Based on the results (b = −0.073, T = 1.328, $p < 0.000$), this study discovered that BG does not modify the relationship between the QIA and the QFR, as the value of (b) is less than 0 and the value of (T) is less than 2, so the fifth hypothesis is rejected.

**Table 7.** Path coefficients.

| STDEV, T Values, *p* Values | | | | |
|---|---|---|---|---|
| Relationship | Beta | T Statistics (\|O/STDEV\|) | *p* Values | Decision |
| BG ⟶ QFR | 0.332 | 6.227 | 0.000 | Supported |
| IA ⟶ QFR | 0.270 | 5.869 | 0.000 | Supported |
| QIA ⟶ QFR | 0.160 | 3.077 | 0.002 | Supported |
| BG × IA ⟶ QFR | −0.048 | 0.868 | 0.385 | Not Supported |
| BG × QIA ⟶ QFR | −0.073 | 1.328 | 0.184 | Not Supported |

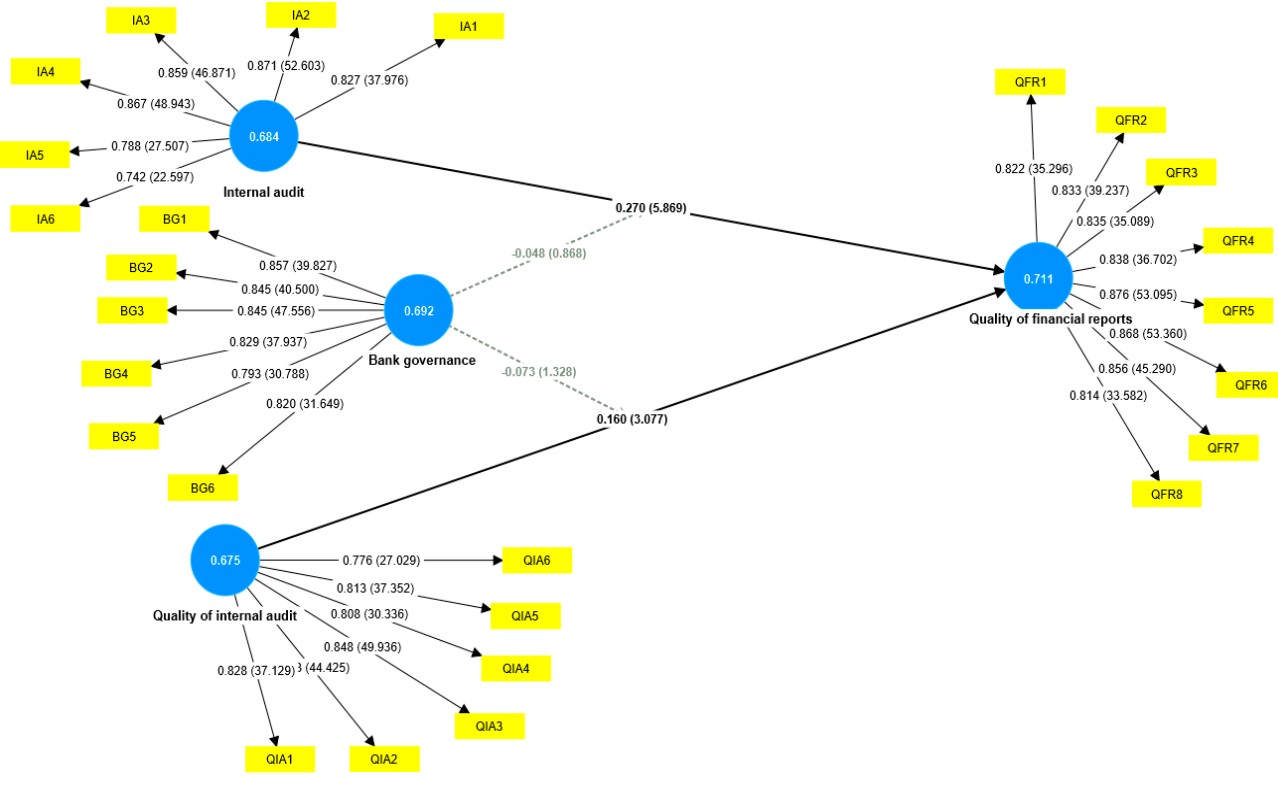

**Figure 4.** The PLS algorithm of the measurement model.

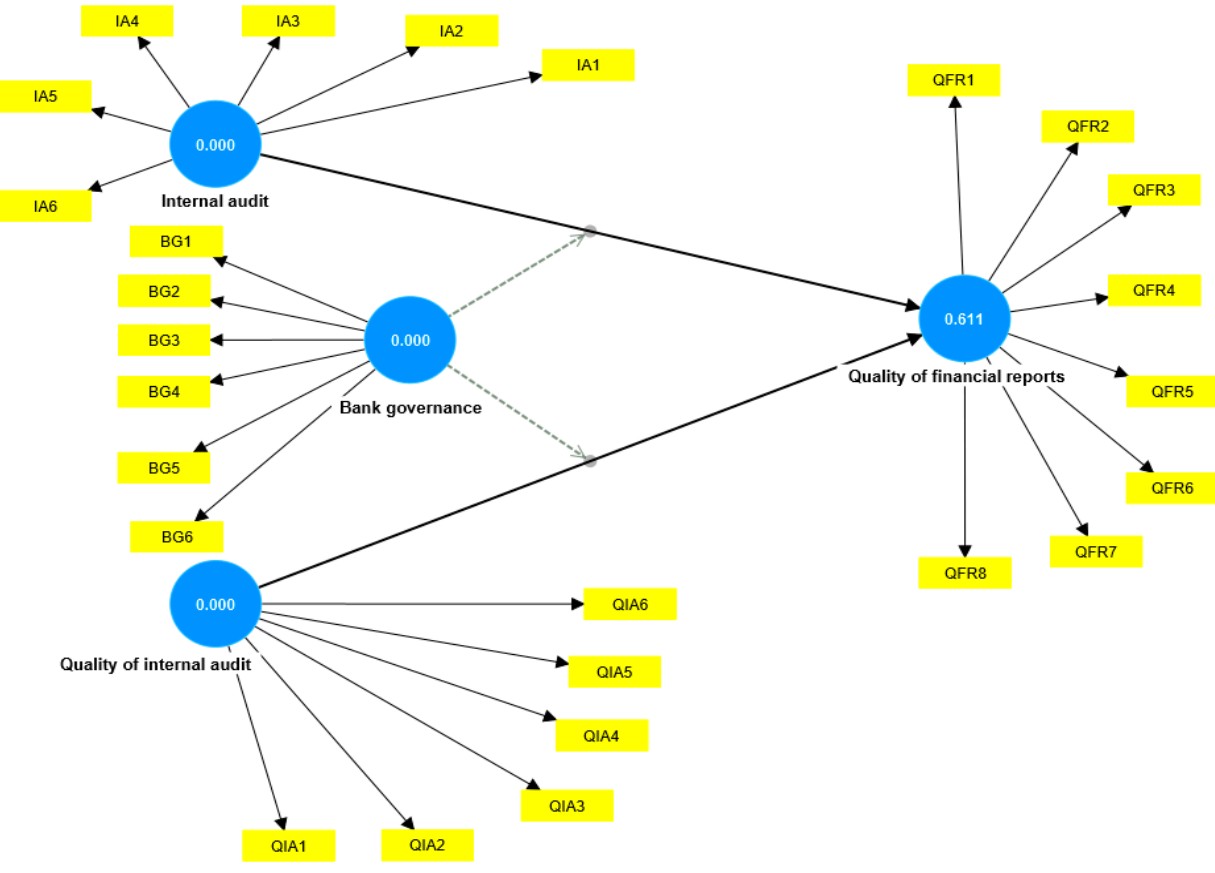

**Figure 5.** The PLS algorithm of the measurement model.

The study delved into the examination of its hypotheses and the establishment of correlations among the research variables through the utilization of the PLS technique and a structural model. The estimated Q2 values were derived from t statistics, path coefficients, and R2. Path coefficients were instrumental in revealing the consistency and direction of correlations, while t statistics and standard errors provided insights into the effect size. The R2 value served as an indicator of the proportion of variation in the dependent variable. The explanatory power of the proposed model was influenced by variations in the dependent variables, and its predictive capacity was assessed by exploring whether the model could accurately predict outcomes using data that were not part of its parameter development. According to the study's findings and R2 values, the suggested framework accounted for approximately 53% of the variation in the effectiveness of the IA and QIA. This suggests that the IA and QIA, as independent variables, could explain 53% of the variation in the QFR score and impact rating. Therefore, the proposed framework is deemed to have a satisfactory level of justification, as suggested by Chin (1998a). Sensitivity analysis through innervation trials with omission yielded second quartile counts comparable to, but significantly higher than, zero and positive values as per Chin (1998b). In this context, the model's explanatory power and predictive utility were considered adequate. The detailed findings are presented in Table 8.

**Table 8.** Values $R^2$, $f^2$, $Q^2$.

| | Error | 2.5% | 97.5% | R | F | Q2 | Decision |
|---|---|---|---|---|---|---|---|
| BG → QFR | −0.001 | 0.224 | 0.435 | | 0.175 | | Supported |
| IA → QFR | 0.001 | 0.177 | 0.359 | | 0.149 | | Supported |
| QIA → QFR | 0.001 | 0.060 | 0.266 | 0.877 | 0.044 | 0.611 | Supported |
| BG × IA → QFR | 0.003 | −0.156 | 0.061 | | 0.004 | | Not Supported |
| BG × QIA → QFR | −0.002 | −0.185 | 0.031 | | 0.008 | | Not Supported |

## 3. Results Desiccations

In this study, the primary objective was to explore the direct influence of the IA and QIA on the QFR. Notably, we investigated whether CG acts as a moderator in shaping the relationships among the IA, QIA, and the enhancement of the QFR within the specific context of the Yemen Central Bank (YCB). The study encompassed the evaluation of the IA, QIA, BG, and QFR. The first hypothesis posited that the IA positively affects the QFR. The study conclusively established that the IA has a positive and significant impact on the QFR, with (b = 0.270, T = 5.869, $p > 0.000$), thereby supporting the acceptance of the hypothesis. The second hypothesis asserted that the QIA positively affects the QFR. The study validated this hypothesis by demonstrating a positive and significant effect of the QIA on the QFR (b = 0.160, T = 3.077, $p > 0.000$). The third hypothesis suggested that BG positively affects the QFR. The study affirmed this hypothesis, revealing a positive and significant relationship between BG and the QFR (b = 0.332, T = 6.227, $p > 0.000$). In contrast, the fourth hypothesis, proposing that BG moderates the relationship between the IA and QFR, was not supported. The study demonstrated that BG does not alter the relationship between the IA and QFR, with (b = −0.048, T = 0.868, $p < 0.000$), leading to the rejection of the hypothesis. Similarly, the fifth hypothesis, asserting that BG moderates the relationship between the QIA and QFR, was not validated. The study revealed that BG does not modify the relationship between the QIA and QFR, with (b = −0.073, T = 1.328, $p < 0.000$), resulting in the rejection of the hypothesis. These findings align with the study conducted by (Hanim Fadzil et al. 2005), supporting the positive and significant relationship between the IA and QFR. Additionally, they echo the conclusions of (Hazami-Ammar 2019), emphasizing the crucial role of the QFR positively correlating with the IA's proactive measures, such as initiating investigations into fraud and irregularities. Moreover, (D'Onza et al. 2015) highlighted a strong and positive relationship between the IA and QIA in terms of value addition, while the study results support the critical role of the QIA in fostering trans-

parency and accountability, ultimately enhancing BG. It is noteworthy that the findings diverge from the results of Oussii and Taktak (2018a), who found no significant relationship between the revealed IA flaws in commercial banks and the independence and impartiality of the IA. These contradictions underscore the complexity and context-specific nature of governance dynamics. In conclusion, the study contributes valuable insights into the nuanced relationships among the IA, QIA, BG, and QFR in the specific context of YCB. The results emphasize the multifaceted impact of the IA and underscore the critical role of the QIA in promoting transparency and accountability, ultimately contributing to enhanced governance performance.

The study marks a pioneering endeavor as the first to seamlessly integrate the IA and QIA. Utilizing the QFR as a dependent variable and BG, the research delves into the examination of the Yemeni banking industry. In the context of the Middle East, this study emerges as one of the few endeavors that scrutinize the state of IA. Within Middle Eastern nations, studies on the QIA remain limited, with a handful of notable works (Oussii and Taktak 2018b). This research seeks to bridge this knowledge gap by providing valuable insights into the practices of the IA in the Republic of Yemen. The evaluation encompasses an in-depth assessment of the effectiveness of the internal audit framework and explores the nuanced relationship between enhancing BG, financial reporting, and the QIA framework. The significance of these findings extends to practitioners, especially Chief Audit Executives (CAEs) aiming to bolster the effectiveness of business operations within their organizations. Internal auditors and CAEs can leverage these insights to identify specific areas for development, aligning their practices with organizational objectives. The research illuminates the internal audit framework components that wield an impact on the QIA. Commercial bank boards and advisory committees stand to benefit from the study's outcomes in executing their duties. This includes the strategic selection of chief audit executives, the recruitment of IA specialists, and the ongoing assessment of the IA system to ensure sustained efficacy. In essence, this study not only breaks new ground in the integration of the IA and QIA, but also contributes actionable insights that can drive improvements in IA practices and governance structures. The contextualized focus on Yemen's economic landscape adds unique value, making the research a significant resource for advancing best practices within the region.

## 4. Conclusions

This study's main aim is to explore the positive impact of the IA and QIA on the QFR. Additionally, we investigate the nuanced relationship between the IA, QIA, and QFR, incorporating BG as a moderator variable. The outcomes of the research contribute to the existing literature, emphasizing the crucial role of the IA and QIA in elevating the QFR. The formulated research hypotheses, with BG as a moderator, seek to understand how the IA and QIA influence the QFR. The empirical results reveal a significant and positive association between the IA and the quality level of financial reports. Notably, the findings underscore a positive relationship between the IA, QIA, and QFR. This underscores the value of the IA and QIA in enhancing the QFR and instilling confidence in the improvement of internal control processes and the financial reporting framework. The study's contributions are manifold. Firstly, it bolsters existing research on the interconnectedness of the IA, QIA, and QFR. It accentuates the pivotal role of the IA, its efficacy, and its capacity to enhance the QFR. This research advocates for stricter internal controls, positing that a strengthened IA and QIA, coupled with improved BG, can empower managers to elevate financial reporting standards in banks. It further provides a mechanism for audit committees to monitor IA processes and assess internal performance.

When incorporating BG as a moderator in the relationship between the IA, QIA, and QFR, the results show that BG as a moderator does not influence the QFR. Moreover, there is no positive relationship observed between the IA, QIA, and QFR. Specifically, the findings of the study suggest that regulators might benefit from reconsidering approaches to enhancing the IA and QIA for improved financial reporting and internal control quality,

fostering a resilient banking system. Acknowledging the study's limitations is crucial for a nuanced interpretation of the results. Firstly, the study's sample size is modest, a common characteristic in the Yemeni market. Secondly, this study focuses on the banking industry and might overlook insights from other Yemeni economic sectors, encouraging future research to explore these areas. Thirdly, the IA and QIA may possess additional characteristics not considered in the study model, urging future investigations into factors like IA risk consulting, senior management support for the IA and QIA, and BG to comprehensively enhance organizational practices and elevate financial reporting standards.

**Author Contributions:** Conceptualization, N.A.M.S.; methodology, N.A.M.S.; software, N.A.M.S.; validation, N.A.M.S.; formal analysis, N.A.M.S.; investigation, N.A.M.S.; resources, N.A.M.S.; data curation, N.A.M.S.; writing—original draft preparation, N.A.M.S.; writing—review and editing, N.A.M.S.; visualization, N.A.M.S.; supervision, N.A.M.S.; project administration, N.A.M.S.; funding acquisition, N.A.M.S. All authors have read and agreed to the published version of the manuscript.

**Funding:** This study is supported via funding from Prince Sattam bin Abdulaziz University project number (PSAU/2024/R/1445).

**Data Availability Statement:** The data can be available with a request from corresponding author.

**Acknowledgments:** During the preparation of this work, the author used Chatgpt 3.5 in order to improve the language. After using this tool, the author reviewed and edited the content as needed and takes full responsibility for the content of the publication.

**Conflicts of Interest:** The author declares no conflict of interest.

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
