# Peer review of "The Moderating Role of Corporate Governance on the Associations of Internal Audit and Its Quality with the Financial Reporting Quality: The Case of Yemeni Banks"

_jrfm, doi:10.3390/jrfm17030124_

Round 1

Reviewer 1 Report

Comments and Suggestions for Authors

Author Response

Comments Responses

Dear respected reviewer.

I appreciate the time and effort that you have dedicated to providing your valuable feedback on my manuscript. I am grateful to you for your insightful comments on my paper. I have been able to incorporate changes to reflect most of the suggestions provided by you. I have highlighted the changes within the manuscript. Here is a point-by-point response to your valuable comments and concerns. As well as my responses attached in a word file.

Reviewer’s Report on “The Moderating Role of Corporate Governance on the Associations of Internal Audit and its Quality with the Financial Reporting Quality: The Case of Yemeni Banks”

This paper examines how corporate governance influences the relationship between internal audit, internal audit quality, and financial reporting quality in Yemeni commercial banks. Surveying 210 participants, including internal auditors, heads of internal audit, and audit committee members, the study finds a positive relationship between internal audit and both internal audit quality and financial reporting quality. However, the moderating effect of corporate governance on this relationship was found to be insignificant, shedding light on the dynamics of financial reporting quality in developing countries such as Yemen.

The article contains novel information, is well-written, and interesting. Nevertheless, I recommend certain improvements to further strengthen it before publication.

Major Comments:

  1. The abstract is well-written and summarizes the paper nicely. However, I would recommend sharpening it up a bit. For example, the last sentence is rather vague: "The results of this study contribute valuable insights to the existing literature on the influence of internal audit, internal audit quality, and corporate governance on financial reporting quality in developing countries, particularly in Yemen."

Response: I have revised the abstract in the paper according to your valuable comment as below:

It is worth noting that results confirm the existence of a positive relationship between internal audit, internal audit quality, and financial reporting quality. This confirms the importance of internal audit and the quality of internal audit in enhancing the quality of financial reports, and instilling confidence in improving internal control processes and the financial reporting framework. Among the study's many contributions is that it enhances current research on the interrelationship between internal auditing, internal audit quality, and financial reporting quality. It highlights the pivotal role of internal audit, its effectiveness and its ability to improve the quality of financial reports. this study calls for more stringent internal controls, and posits that strengthening internal audit and internal audit quality, along with improving corporate governance, can enable managers to raise financial reporting standards in banks. It also provides a mechanism for audit committees to monitor internal audit processes and evaluate internal performance

  1. The introduction needs some work and could be improved in several ways. The first two paragraphs are fine. In the third, the authors begin by stating what the authors are doing, but never finish, limiting themselves to stating the aim. The authors should go on to summarize the results so that the reader can understand what the paper is about. Then the authors return to summarizing the literature again, starting with "Accounting fraud is a widespread global problem, with...". If this is going to be the summary of contributions, then why not state it directly, along the lines of "Our study contributes to X areas of research, First, ..... etc. Finally, it would be useful to end the introduction with a short paragraph outlining the structure of the paper.

Response: Thank you for pointing this out. I agree with this comment. Therefore, I have amended, summarized the results, and ended the introduction with a short paragraph outlining the structure of the paper as required by the above comment as follow:

The study results confirm the existence of a positive relationship between internal audit, internal audit quality, and financial reporting quality. This confirms the importance of internal audit and the quality of internal audit in enhancing the quality of financial reports, and improving internal control processes and the financial reporting framework. It highlights the pivotal role of internal audit, its effectiveness and its ability to improve the quality of financial reports. It is assumed that strengthening internal audit and internal audit quality, along with improving corporate governance, can enable managers to raise financial reporting standards in banks. It also provides a mechanism for audit committees to monitor internal audit operations and evaluate internal performance.

Financial reports are of great importance in the financial market and financial companies, as reports indicate that a large percentage of organizations fail to meet the quality standards required in their financial reports. A PricewaterhouseCoopers survey in 2018 found that most reports did not meet the necessary standards.

Based on the above, the current study examines The moderating role of corporate governance on the associations of internal audit and its quality with the financial reporting quality: The case of Yemeni banks.

The problem of the study: What is the impact of corporate governance on internal audit engagements and their quality on the quality of financial reports in the Commercial Bank of Yemen?

  1. The authors focus on Yemen, which is quite interesting. There are not many studies on this topic. But why should we focus on Yemen? Are there any unique insights we can learn from studying this country? Any unique institutional or cultural settings? This would strengthen the motivation and could be mentioned in the introduction.

Response: Thank you for this suggestion. It would have been interesting to explore this aspect. therefore, in the case of this study which focused on Yemen because Yemen is one of the developing countries which need a lot of work to be done specially in research field as well as the researcher is from the Republic of Yemen.

  1. After reading the literature review section, I feel that the different subsections are largely disconnected. The authors could work more on the narrative to improve the flow. Also, it would be good to be more convincing about what the purpose of this section is. Do the authors want to identify the gap or develop the hypotheses? If so, perhaps it would be good to spell out all the hypotheses, not just H4 and H5 as selected.

Response: I agree with this and have incorporated your suggestion by identifying the study gap after reviewing the literature and the study problem after the introduction as well as, the hypotheses H1, H2, and H3 were spelled out.

Through previous studies that did not talk about the moderating effect of corporate governance on the relationship between internal auditing and its quality with the quality of financial reports in the Commercial Bank of Yemen. There were no previous studies at all that indicated this, and therefore the following study gap was formulated. Trying to know the direct and indirect impact of internal auditing and its quality on the quality of financial reports in the Commercial Bank of Yemen when using a modified corporate governance variable.

  1. In addition to the above, I would be interested to see a discussion of the role of Islamic finance in the study. Islamic banking is an important factor in the Yemeni landscape, but I find it rather neglected. It has now been demonstrated that adherence to Islamic principles has a significant impact on bank performance. While I do not ask the authors to necessarily include it in the empirical analysis, it would be beneficial to at least acknowledge this fact by referring to the relevant literature (see, for example, (Albaity et al., 2019; Khan et al., 2023).

Response: Thank you for this suggestion. It would have been interesting to explore this aspect. However, in the case of our study, it seems slightly out of scope because this study based on profitable commercial banks therefore it is better not to discuss the Islamic banks in this paper. While the Islamic finance is very important in all Muslim communities including Yemen therefore I have a separate study related to that with their standards and relevant literature.

Minor Comments:

  1. I suggest shortening the title, as shorter titles tend to attract more citations. My tentative suggestion is "Corporate governance, internal audit, and financial reporting quality: The case of Yemeni banks." It would also be beneficial to mention that the study is based on a unique sample from frontier markets.

Response: Thank you for this suggestion. It would have been interesting to explore this aspect. However, in the case of our study, I think the title is more appropriate as it is.

  1. Tables are not self-contained. As a result, the reader sometimes needs to scan the entire article to properly understand the content. Authors should ensure that the notes to the tables and figures provide sufficient explanation of their content, symbols, etc.

Response: Thank you for this suggestion. It would have been interesting to explore this aspect. Therefore, each table has included with a detailed explanation throughout the manuscript.

References:

Albaity, M., Mallek, R. S., & Noman, A. H. M. (2019). Competition and bank stability in the MENA region: The moderating effect of Islamic versus conventional banks. Emerging Markets Review, 38, 310-325.

Khan, M. F., Ali, M. S., Hossain, M. N., & Bairagi, M. (2023). Determinants of nonperforming loans in conventional and Islamic banks: Emerging market evidence. Modern Finance, 1(1), 56–69. https://doi.org/10.61351/mf.v1i1.27

Reviewer 2 Report

Comments and Suggestions for Authors

The paper examine the impact of II and QIA on QFR for Yemen Banks. There are some overlaps in the measures of IA and QIA along with the issues of measurement errors and the importance weights on each survey items.

The proposed hypothesis H4  and H5 are about the impact of BG through IA and QIA to QFR, which are rejected. For me this seems a natural result if  the effect of BG is already reflected into IA and QIA. And authors need to check this possibility or the result on H4 & H5. 

Comments on the Quality of English Language

Writing in English seems fine.

Author Response

Comments Responses

Dear respected reviewer.

I appreciate the time and effort that you have dedicated to providing your valuable feedback on my manuscript. I am grateful to you for your insightful comments on my paper. I have been able to incorporate changes to reflect most of the suggestions provided by you. I have highlighted the changes within the manuscript. Here is a point-by-point response to your valuable comments and concerns. As well as my responses attached in a word file.

The paper examines the impact of II and QIA on QFR for Yemen Banks. There are some overlaps in the measures of IA and QIA along with the issues of measurement errors and the importance weights on each survey items.

Response: Thank you for pointing this out. Therefore, I have checked all measurements and found them appropriate.

The proposed hypothesis H4 and H5 are about the impact of BG through IA and QIA to QFR, which are rejected. For me this seems a natural result if the effect of BG is already reflected into IA and QIA. And authors need to check this possibility or the result on H4 & H5. 

Response: Thank you for pointing this out. I agree with this comment. Therefore, I checked the result on the fourth and fifth hypotheses which are rejected because the T in the two hypotheses is less than 2 and the P value is greater than 0.05.

Reviewer 3 Report

Comments and Suggestions for Authors

The introduction needs to be completed and clarified regarding the GAP phenomenon in financial reporting quality: The case of Yemeni banks.

In the introduction to the 4th paragraph, you can focus more on financial reporting quality; don't focus on financial reporting fraud.

The research gap explanation regarding and why testing BG moderates the relationship between QIA and QFR must be explained in more detail.

The novelty and formulation of the research problem need to be explained clearly at the end of the introduction.

In the Literature Review section, explaining the theory used regarding the variables studied is better.

In the hypothesis section, for example, BG moderates the relationship between IA and QFR, which must be explained along with the supporting theory.

The methodology must be equipped with sampling tables and operational definitions of variables.

Statements H1, H2, and H3 have yet to be explained clearly in the hypothesis; please explain in detail.

Explain each hypothesis result individually, especially H4 and H5, which is necessary for the discussion.

In the conclusion, it is necessary to explain the implications of the research and suggestions for further study.

For references, it is best to use the last five years; several references need to be completed regarding the journal name and DOI. Such as the following:

Bartov, E., Gul, F. A., & Tsui, J. S. (2000). Discretionary-accruals models and audit qualifications. Journal of Accounting and Economics, 30(3), 421-452

BCBS. (2012). The internal audit function in banks. https://www.bis.org/publ/bcbs223.htm

Blackburn, R., & Jarvis, R. (2010). The role of small and medium practices in providing business support to small and medium-sized enterprises

Burton, F. G., Emett, S. A., Simon, C. A., & Wood, D. A. (2012). Corporate managers' reliance on internal auditor recommendations. Auditing: A Journal of Practice & Theory, 31(2), 151-166

Field, A. (2009). Discovery Statistics Using SPSS: sex, drugs and Rock 'n' Roll

Hanim Fadzil, F., Haron, H., & Jantan, M. (2005). Internal auditing practices and internal control systems. Managerial Auditing Journal, 20(8), 844-866.

Nkundabanyanga, S. K., & Ahiauzu, A. (2012). Board role performance in Uganda's services sector firms. Nkundabanyanga, S. K., & Ahiauzu, A. (2012). Board role performance in Uganda's services sector firms.

Author Response

Comments Responses

Dear respected reviewer.

I appreciate the time and effort that you have dedicated to providing your valuable feedback on my manuscript. I am grateful to you for your insightful comments on my paper. I have been able to incorporate changes to reflect most of the suggestions provided by you. I have highlighted the changes within the manuscript. Here is a point-by-point response to your valuable comments and concerns. As well as my responses attached in a word file.

The introduction needs to be completed and clarified regarding the GAP phenomenon in financial reporting quality: The case of Yemeni banks.

Response: Thank you for pointing this out. I agree with this comment. Therefore, I have clarified the GAP phenomenon in financial reporting quality: The case of Yemeni banks.

In the introduction to the 4th paragraph, you can focus more on financial reporting quality; don't focus on financial reporting fraud.

Response: Thank you for pointing this out. I agree with this comment. Therefore, I have amended and modified by focusing on financial reporting quality.

An explanation of the research gap regarding and why BG testing moderates the relationship between QIA and QFR should be explained in more detail.

Response: I agree with this and have incorporated your suggestion by formulating and explaining the relationship between QIA and QFR and the moderating role of BG in more details.

The novelty and formulation of the research problem need to be explained clearly at the end of the introduction.

Response: I agree with this and have incorporated your suggestion through formulating the problem of the study at the end of the introduction.

In the Literature Review section, explaining the theory used regarding the variables studied is better.

Response: Thank you for this suggestion. It would have been interesting to explore this aspect. However, In the literature review section, it is best to explain the theory used in relation to the variables studied.

In the hypothesis section, for example, BG moderates the relationship between IA and QFR, which must be explained along with the supporting theory.

The methodology must be equipped with sampling tables and operational definitions of variables.

Response: Thank you for pointing this out. I agree with this comment. Therefore, I have formulated table 1 in which the study sample and the number of respondents in each bank were determined.

Statements H1, H2, and H3 have yet to be explained clearly in the hypothesis; please explain in detail.

Explain each hypothesis result individually, especially H4 and H5, which is necessary for the discussion.

Response: Thank you for this suggestion. It would have been interesting to explore this aspect. However, the statements of hypotheses were explained in detail. As well as each hypothesis result were discussed individually.

In the conclusion, it is necessary to explain the implications of the research and suggestions for further study.

Response: Thank you for pointing this out. I agree with this comment. Therefore, I have formulated and added the study suggestions in the conclusion.

 For references, it is best to use the last five years; several references need to be completed regarding the journal name and DOI. Such as the following:

Bartov, E., Gul, F. A., & Tsui, J. S. (2000). Discretionary-accruals models and audit qualifications. Journal of Accounting and Economics, 30(3), 421-452

BCBS. (2012). The internal audit function in banks. https://www.bis.org/publ/bcbs223.htm

Blackburn, R., & Jarvis, R. (2010). The role of small and medium practices in providing business support to small and medium-sized enterprises

Burton, F. G., Emett, S. A., Simon, C. A., & Wood, D. A. (2012). Corporate managers' reliance on internal auditor recommendations. Auditing: A Journal of Practice & Theory, 31(2), 151-166

Field, A. (2009). Discovery Statistics Using SPSS: sex, drugs and Rock 'n' Roll

Hanim Fadzil, F., Haron, H., & Jantan, M. (2005). Internal auditing practices and internal control systems. Managerial Auditing Journal, 20(8), 844-866.

Nkundabanyanga, S. K., & Ahiauzu, A. (2012). Board role performance in Uganda's services sector firms. Nkundabanyanga, S. K., & Ahiauzu, A. (2012). Board role performance in Uganda's services sector firms.
